# Assessing BRCA1 activity in DNA damage repair using human induced pluripotent stem cells as an approach to assist classification of BRCA1 variants of uncertain significance

**Meryem Ozgencil**[1¤], **Julian Barwell**[2], **Marc Tischkowitz**[3], **Louise Izatt**[4], **Ian Kesterton**[5], **Michael Simpson**[1], **Paul Sharpe**[6], **Paulo de Sepulveda**[7], **Edwige Voisset**[1‡*], **Ellen Solomon**[1‡*]

1 Department of Medical & Molecular Genetics, King's College London, Faculty of Life Sciences & Medicine, London, United Kingdom, 2 Department of Genetics and Genome Biology at the University of Leicester, Leicester, United Kingdom, 3 Department of Medical Genetics, National Institute for Health Research Cambridge Biomedical Research Centre, University of Cambridge, Cambridge, United Kingdom, 4 Clinical Genetics, Guy's and St Thomas' NHS Foundation Trust, London, United Kingdom, 5 Cytogenetics Laboratory, Viapath Analytics, Guy's and St. Thomas' NHS Foundation Trust, Guy's Hospital, London, United Kingdom, 6 Department of Craniofacial Development & Stem Cell Biology, King's College London, London, United Kingdom, 7 Signaling Hematopoiesis and Mechanism of Oncogenesis Lab, INSERM, CNRS, Institut Paoli-Calmettes, CRCM, Aix Marseille University, Marseille, France

¤ Current address: Centre for Cancer Cell & Molecular Biology, Barts Cancer Institute, Queen Mary University of London, John Vane Centre, Charterhouse Square, London, United Kingdom
‡ EV and ES are Co-last authors on this work.
* edwige.voisset@inserm.fr (EV); ellen.solomon@kcl.ac.uk (ES)

**Data Availability Statement:** All relevant data are within the paper and its Supporting Information files.

## Abstract

Establishing a universally applicable protocol to assess the impact of BRCA1 variants of uncertain significance (VUS) expression is a problem which has yet to be resolved despite major progresses have been made. The numerous difficulties which must be overcome include the choices of cellular models and functional assays. We hypothesised that the use of induced pluripotent stem (iPS) cells might facilitate the standardisation of protocols for classification, and could better model the disease process. We generated eight iPS cell lines from patient samples expressing either BRCA1 pathogenic variants, non-pathogenic variants, or BRCA1 VUSs. The impact of these variants on DNA damage repair was examined using a H2AX foci formation assay, a Homologous Repair (HR) reporter assay, and a chromosome abnormality assay. Finally, all lines were tested for their ability to differentiate into mammary lineages in vitro. While the results obtained from the two BRCA1 pathogenic variants were consistent with published data, some other variants exhibited differences. The most striking of these was the BRCA1 variant Y856H (classified as benign), which was unexpectedly found to present a faulty HR repair pathway, a finding linked to the presence of an additional variant in the ATM gene. Finally, all lines were able to differentiate first into mammospheres, and then into more advanced mammary lineages expressing luminal- or basal-specific markers. This study stresses that BRCA1 genetic analysis alone is insufficient to establish a reliable and functional classification for assessment of clinical risk, and that it cannot be performed without considering the other genetic aberrations which may be

**Funding:** This work was supported by a European Union Scholarship Programme, by a pilot grant from Breast Cancer Now (2015NovPR609), by a specialist program grant from Blood Cancer UK (13043; previously called Bloodwise) and King's College London. All BRC facilities were funded by the National Institute for Health Research (NIHR) Biomedical Research Centre, based at Guy's and St. Thomas' NHS Foundation Trust and King's College London. The funders had no role in study design, data collection and analysis, decision to publish, or preparation of the manuscript.

**Competing interests:** The authors have declared that no competing interests exist.

present in patients. The study also provides promising opportunities for elucidating the physiopathology and clinical evolution of breast cancer, by using iPS cells.

## Introduction

Worldwide, among females, breast cancer is the most commonly diagnosed cancer ($\sim$24%), and is still the leading cause of cancer-related deaths in women, closely followed by lung cancer (respectively 15.5% and 13.7%; [1, 2]). The breast cancers form a highly heterogeneous group, and are consequently clinically challenging to diagnose and manage. Variants in the genes BRCA1 or BRCA2 are known to confer a high lifetime risk of developing breast cancer, because the presence of certain heterozygous BRCA variants increases the risk of breast (~80% for BRCA1) and ovarian cancer (~40% for BRCA1), identifying BRCA1 and BRCA2 as high-penetrance breast cancer predisposition genes [3]. Additionally, BRCA1 and BRCA2 are the most common genes associated with increased risk in hereditary breast and ovarian cancer.

BRCA1 is a tumour suppressor gene coding for a large protein containing multiple functional domains, which interacts with multiple other proteins [4, 5]. The BRCA1 gene is an essential component of DNA damage repair via the Homologous Recombination (HR) pathway, which is an error-free repair mechanism, and hence crucial for cellular survival. BRCA1 is also involved in chromosome segregation and mismatch repair, so that it also plays a critical role in the maintenance of genome integrity [6].

While some BRCA1 variants are undoubtedly pathogenic, over 50% of BRCA1 variants (single nucleotide variants) are classified as Variants of Uncertain Significance (VUSs) on the ClinVar public database, thus complicating clinical decisions regarding the choice of therapies and counselling for reproductive decisions. There are currently >5200 distinct germline variants of BRCA1 listed on ClinVar, including >1300 VUSs, and an additional 175 variants are subject to conflicting interpretations [7]. There is, therefore, an imperative need to improve the evaluation and classification of BRCA1 VUSs.

To achieve this, various functional assays have been assessed, in order to make an accurate evaluation of the risk posed by newly identified BRCA1 variants, and also to address the issue of BRCA1 haploinsufficiency. Unfortunately, some of these assessments have been problematic, chiefly because of the variety of experimental models (e.g. lymphocytes/lymphoblastic cell lines) and protocols used. Based on the observations in these studies, attempts were then made to standardise the evaluation of VUSs [8].

In this study, we sought to investigate an alternative state-of-the-art approach using induced Pluripotent Stem cells (iPS). iPS cells are a useful tool for disease modelling with the advantage, compared to embryonic stem cells, that they already contain the specific variant [8–12]. For the purposes of this study, iPS cells have been derived from carriers of pathogenic variants and VUSs, and functional assays have been carried out to assess the efficiency of DNA damage repair.

## Materials and methods

### Ethics approval and consent to participate

Either study participants agreed and signed a written informed consent for inclusion in the KHP CANCER BIOBANK (HTA Licence No: 12121, REC No: 12-EE-0493), or patients were recruited using the Investigating Hereditary Cancer Predisposition–a combined genomics, approach study that was approved by the NRES Committee East of England–Hertfordshire (Ethics Ref: 12/EE/0478: IRAS 114545: EDGE 17857).

## Generation of iPS cells from heterozygous BRCA1 variant carriers

Fibroblasts were derived from skin biopsies, grown and expanded in DMEM with 10% foetal bovine serum (FBS) and 5% penicillin/streptomycin (ThermoFischer Scientific). Control and carrier fibroblasts were reprogrammed using CytoTune I or II Sendai Viral Transduction (ThermoFischer Scientific), following the manufacturer's instructions. On day 7 reprogrammed cells were been transferred onto NuFF feeders, with either Pluriton (Stemgent) or Nutristem (Stemgent) used to support their growth. Cells were also adapted to be feeder-free, and maintained on 0.34 μg/μl GFR Matrigel with TeSR E8 (StemCell Technologies). All the derived lines were confirmed by qPCR to be free from exogenous Sendai viral factors using the TaqMAn iPSC Sendai Detection Kit (ThermoFischer Scientific) (S1A–S1I Fig).

## Pluripotency staining of iPS cells

Pluripotency staining was performed according to the protocol of Ilic et al. (2012) [13]. Briefly, cells were fixed with 4% PFA and permeabilised with 0.1% Triton-X for 10 minutes. Blocking was carried out for 1 hour with 5% BSA in PBS-Tween. Incubation with TRA-1-60, TRA1-81 (Millipore), NANOG (R&D Systems) and OCT4 (Santa Cruz Technologies) antibodies was carried out overnight at 4˚C, followed by incubation with an appropriate secondary antibody for 1 hour at room temperature. Cells were mounted in vectashield mounting medium containing DAPI, and visualised using a Nikon 50i epifluorescence microscope.

## Pluripotency assays

For in vivo differentiation of iPS Cells: $1\times10^6$ cells were resuspended in 50 μL 1:3 diluted GFR matrigel (BD Biosciences). The cell suspension was then injected into the left testis of a severe combined immunodeficiency (SCID) mouse, while the right testis was injected with 1:3 diluted GFR Matrigel alone, as a negative control. Following teratoma formation, classic histological staining was carried out using Mayer's Haematoxylin and Eosin. Images were taken using a Zeiss Axioscope Z Plus microscope.

For in vitro differentiation of iPS cells: iPS cells were grown for 10 days in DMEM supplemented with 20% FBS. Medium was changed once every three days, and cells were then fixed with 4% PFA. The following antibodies were used: anti-α-fetoprotein (endoderm), anti-βIII-tubulin (ectoderm), and anti-α-smooth muscle actin (mesoderm), all from Millipore. Images were taken using a Nikon Eclipse 50i microscope at x10 magnification.

## Array comparative genomic hybridization (aCGH)

1 μg of DNA was labelled using a CGH labelling kit (Enzo Life Sciences), purified using the QIAquick PCR purification kit (Qiagen), and run on an Agilent 4 x 44 K platform using either Wessex NGRL design 017457 or design 028469. Hybridization, washing and scanning of the arrays were all carried out according to the manufacturer's protocols.

## Whole exome sequencing

DNA was extracted using QIAamp DNA Extraction kit (Qiagen). For each line a DNA concentration of 30 ng/μl was used. Libraries were prepared using SureSelect Human All Exon Capture v4 (Agilent), and sequenced with 100bp paired end reads on HiSeq platform (Illumina), at the Biomedical Research Centre (BRC) Genomics Facility. Reads were aligned to the human reference genome (h37) using NovoAlign (Novocraft), and somatic mutations were called using VarScan2 [14] with alignments in Samtools pileup format [15]. Annotations were then performed using ANNOVAR [16].

## Sanger sequencing

Sequencing was carried out using a Big Dye Terminator v3.1 cycle sequencing kit (Applied Biosystems). Sequencing was carried out on the ABI 3730xl 96 capillary sequencer. All primers used are listed in S1 Table.

## Plasmid constructs and transfection

Mutations were introduced into full-length WT-BRCA1 pEGFP-N2 construct [4] by using Q5 Site-Directed Mutagenesis kit (NEB) following manufacturer's protocol. Transfections of WT and mutated BRCA1 constructs were carried out by using ProFection mammalian transfection system (Promega) following manufacturer's guidelines.

## Western blot analysis

iPS cells were lysed in SDS lysis buffer containing a protease inhibitor cocktail (Roche). Lysates were incubated at 100˚C for 10 minutes, and sonicated using a Qsonica Water Bath Sonicator. Western blots were then performed as previously described [17]. Antibodies used were as follows: anti-BRCA1 (D9 from Santa Cruz Technologies or Cell Signalling Technology), anti-GFP (Roche), anti-Phospho ATM Ser1981 (Millipore), ATM (Cell Signalling) and anti-HSP90 (Enzo Life Sciences).

## Chromosomal abnormality assay

iPS cells were irradiated with 1 Gy of ionising radiation (IR), using a gamma cell 1000 Elite irradiator (137Cs source). Thirty minutes after IR, 0.1 μg/mL colcemid (ThermoFischer) was added to all samples, and 120 minutes after IR cells were collected. Hypotonic Solution supplemented with 10 μM Rock inhibitor was added, and the hypotonic treatment was performed for 10 minutes at room temperature. A drop of fix (3:1 methanol:glacial acetic acid) was added, and the solution was mixed. Cells were centrifuged at 1,100 rpm for 5 minutes. Supernatant was discarded and 5 mL of fix was added to the cells. Cells were centrifuged, and the fixation step was repeated three times. The metaphase cells were then processed following standard cytogenetic procedures, and chromosomal breaks were scored in >30 metaphase cells, from three independent experiments, for each group.

## HR reporter assay

An HR reporter construct was obtained from the group of Vera Gorbunova, (University of Rochester) [18]. The construct was digested with I-Sce enzyme (NEB), purified from agarose gel using a combination of Freeze 'N Squeeze (Bio-Rad) and a QIAEXII Gel Extraction kit (Qiagen). Nucleofection of iPS cells was carried out according to the protocol of Yang et al (2014) [19]. In brief, 5x10$^5$ iPS cells were transfected with 2 μg linearised HR reporter construct and 5 μg pDsRed-Express-N1 plasmid (Clontech; as transfection efficiency control) by electroporation, using the Amaxa 4D-Nucleofector platform (program CB-150) and a P3 Primary Cell kit with nucleocuvette strip (Lonza) as per the manufacturer's instructions. Cells were recovered in one well of a 24-well plate, and grown for 72 hours. Flow cytometry was performed on a BD LSRFortessa cell analyzer, and data were analyzed with FlowJo software (TreeStar).

## Viability assay

iPS cells have been treated with 1 Gy of irradiation and seeded onto 96 well plates coated with Matrigel. Cellular viability was assessed up to three days after the treatment by using Alamar Blue (Thermo Fisher Scientific) following the manufacturer's protocol.

## γH2AX foci analysis with Amnis ImageStream$^X$ Mk II

At various time points after irradiation (1 Gy), cells were fixed with 4% PFA, and washes were carried out using PBS containing 10% FBS and 0.02M EDTA. Samples were then permeabilised with 0.1% Triton-X for 10 minutes, and blocked with PBS containing 10% FBS and 0.02M EDTA for 1 hour. Following incubation with FITC anti-γH2AX S139 antibody (Biolegend), samples were resuspended in 30 μl PBS containing 0.02 M EDTA and DAPI (2 μg/ml). Stained cells were analysed using an ImageStream$^X$ Mark II imaging flow cytometer, and all images were captured at 40x magnification.

Firstly, single, focused and DAPI positive cells were selected. In order to analyse γH2AX foci in the nucleus, the mask Object (M07, DAPI, Tight) was first applied. To permit more in-depth analysis, a second mask of features was applied [Spot ((Object (M07, DAPI, Tight)), γH2AX Bright, $n_1$,$n_2$) AND Peak (M02, γH2AX, bright, $n_3$)], where $n_1$ represents spot to cell background ratio, $n_2$ represents the radius of the spot, and $n_3$ represents the brightness of the spot. This second mask allows the analysis of γH2AX foci, with specific size and brightness compared to background. An average of 10,000 cells were analysed.

## Mammosphere formation

iPS cells were dissociated into single cells, and 30,000 cells were seeded onto each well of an ultra-low adherent suspension 6-well plate (Corning). Cells were cultured with Mammocult media containing 4 μg/mL Heparin and 0,48 μg/mL Hydrocortisone (StemCell Technologies). Cells were subcultured for up to three passages. Suspension cells were then collected, and centrifuged at 350 g for 5 min, before spheres were dissociated using Trypsin (Thermofisher). Cells were then passed through a 1 ½ Gauge needle to improve dissociation. Hank's Balanced Salt Solution (Sigma), supplemented with 2% FBS (Thermofisher), was added, and cells were centrifuged at 350 g for 5 min. Supernatant was removed, and cells were resuspended in Mammocult media, then seeded at a density of $2x10^4$ per mL in ultra-low attachment 6-well plates (Corning).

## Mammosphere luminal and basal differentiation

Cells were collected in Mammary Epithelial Cell Growth Medium (MEBM) containing 2.5 μg hEGF, 0.25 mg hydrocortisone, 2.5 mg insulin, and 35 mg bovine pituitary extract (BPE). Mammospheres were dissociated, and resuspended in MEBM containing 2.5 μg hEGF, 0.25 mg hydrocortisone, 2.5 mg insulin, and 35 mg BPE (Lonza). Cells were seeded at a density of 30,000 cells per well on bovine Collagen I (ThermoFischer)-coated 24 well plates, and kept in culture for up to 6 days before fixation and staining.

# Results

## Generation of iPS cell lines from BRCA1 variant carriers

Skin samples were obtained from pathogenic BRCA1 variant carriers and VUS BRCA1 carriers undergoing either risk-reducing mastectomy (K381X, A1708E) or tumour-removal surgery (C61G, G462R, Y856H, D1733G, Q1811K, V1687G) (Fig 1A and 1B). The family history of each individual with a VUS, including cancer type and age of onset where available, are

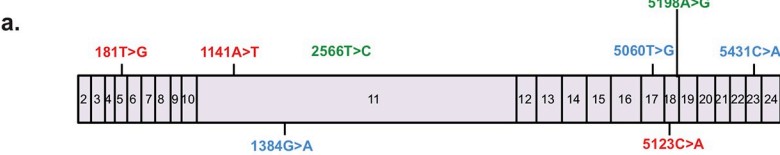

a.

b.

| Nucleotide change | Amino acid change | Type of mutation | ClinVar designation | Cancer history | ClinVar individuals | |
|---|---|---|---|---|---|---|
| 181T>G | C61G | Missense | Pathogenic | Breast cancer | 330 | Ring NLS NLS DBD SCD BRCT BRCT (C61G) |
| 1141A>T | K381X | Nonsense (stop) | Pathogenic | None | 1 | Ring NLS NLS (K381X) |
| 5123C>A | A1708E | Missense | Pathogenic | None | 82 | Ring NLS NLS DBD SCD BRCT BRCT (A1708E) |
| 2566T>C | Y856H | Missense | Benign | Breast cancer | 20 | Ring NLS NLS DBD SCD BRCT BRCT (Y856H) |
| 5198A>G | D1733G | Missense | Likely benign | Breast cancer | 7 | Ring NLS NLS DBD SCD BRCT BRCT (D1733G) |
| 1384G>A | G462R | Missense | VUS | Breast cancer | 3 | Ring NLS NLS DBD SCD BRCT BRCT (G462R) |
| 5060T>G | V1687G | Missense | VUS | Breast cancer | 0 | Ring NLS NLS DBD SCD BRCT BRCT (V1687G) |
| 5431C>A | Q1811K | Missense | VUS | Ovarian cancer | 0 | Ring NLS NLS DBD SCD BRCT BRCT (Q1811K) |

**Fig 1. BRCA1 variant carrier fibroblasts collected and reprogrammed into iPS cells. a** Schematic representation of the human BRCA1 mRNA, with pathogenic variants labelled in red, non-pathogenic labelled in green, and VUSs labelled in blue. BRCA1 exons are numbered from 2 to 24. **b** Table summarising the BRCA1 status of the study subjects, with schematic representations of the human BRCA1 protein variants. RING: Really Interesting New Gene; NLS: Nuclear Localisation Signal; DBD: DNA Binding Domain; SCD: Serine Containing Domain; BRCT: BRCA1 C-Terminus. Data shown are up to date as of January, 2020.

described in S2A–S2E Fig. Three pathogenic [20, 21], two non-pathogenic and three VUS variant-carrying fibroblasts were reprogrammed using Sendai viral transduction, and the formation of iPS cell colonies was observed within a 10-15-day window (Fig 2A and 2B). Non-BRCA1 variant control iPS cells were generated by reprogramming fibroblasts obtained from St John's Institute of Dermatology and the HipSci Biobank (http://www.hipsci.org/). None of the genotypes examined exhibited significant changes in iPS cell colony frequency (colonies per number of cells infected) (S3 Fig).

## Characterisation of *BRCA1* variant iPS cells

Following isolation of at least four independent colonies for each genotype, the expression of conventional pluripotency markers (OCT3/4, NANOG, TRA-1-60 and TRA-1-81) was confirmed by immunofluorescence (Fig 2C, S4–S11 Figs). To further demonstrate pluripotency, iPS cells were injected into the testes of severe combined immunodeficiency (SCID) mice: the four lines tested (iC61G_2, iK381X_1, iY856H_2 and iWT1_2) successfully formed teratomas without any differences in their differentiation potential (Fig 2D, S4–S6 Figs). The six other lines were examined in vitro for three well-established germ layer markers: smooth muscle

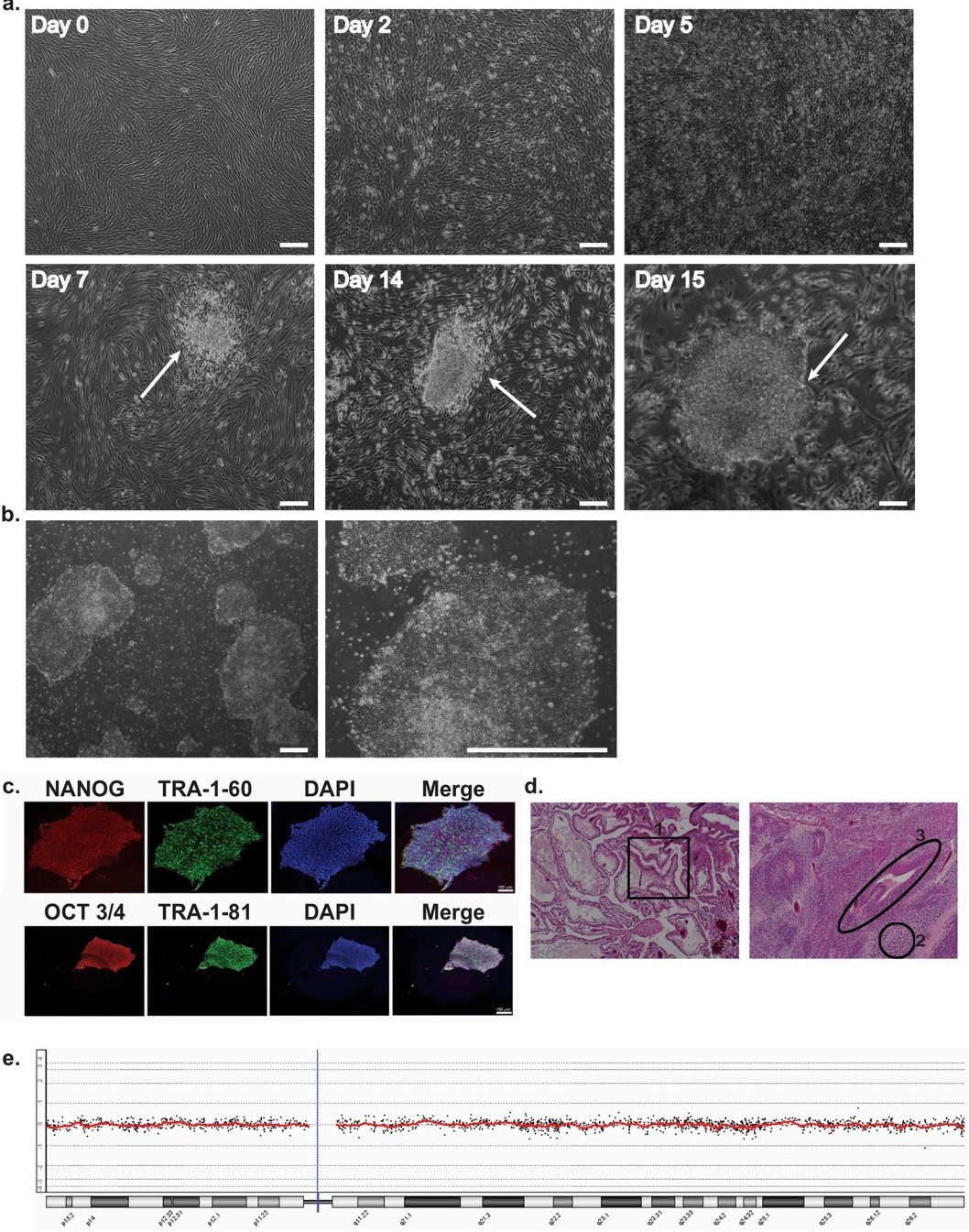

**Fig 2. Derivation of iPS cells from fibroblasts and characterisation (example of iC61G_2). a** Representative brightfield images of reprogramming cells at the indicated time points during iPS cells reprogramming process. Emergence of the first colonies was observed around day 7. White arrows show colonies. **b** Colonies adapted feeder free. Images were taken using an Olympus ix50 microscope at 4x and 10x magnifications. Scale bars represent 250 μm. **c** Representative staining for pluripotency markers. Nuclei were counterstained with DAPI (blue). Images were taken using an Eclipse 50i upright epifluorescence microscope at 50x magnification. Scale bars represent 100 μm. **d** Representative histological analysis of hematoxylin-eosin-stained images of sections of teratomas derived from iPS cells, showing all three germ-layers labelled (1 = endoderm, 2 = mesoderm, 3 = ectoderm). Images were taken using a Zeiss Axioscope Z plus at x10 magnification. **e** Representative aCGH graph.

actin (SMA), β-III Tubulin, and α-Fetoprotein (S7–S11 Figs) [22]. Again, all lines were able to differentiate without any obvious variations in their differentiation patterns. Taken together, these results demonstrate that the iPS cell lines we generated are pluripotent, and contribute to the development of all three germ layers: ectoderm (neural tube), endoderm (gut epithelium), and mesoderm (cartilage). Additionally, all lines maintained genomic stability up to passage P15, as demonstrated by the absence of significant abnormality in aCGH, and neither the wild-type or the variant allele of BRCA1 was lost during the reprogramming process (Fig 2E). These results have been confirmed by whole exome sequencing (WES) of fibroblast samples, and their iPS cell lines derived, as no additional non-synonymous single nucleotide variants (SNVs) predicted to be deleterious were acquired during reprogramming (S1 File) [23].

To examine the impact of BRCA1 variants on BRCA1 protein expression in each heterozygous variant line, Western blot analyses were conducted (Fig 3A and 3B). As expected, a full-length BRCA1 protein was detected at a similar expression level in all pathogenic, non-pathogenic and VUS iPS cell lines, compared to iWT1_2 and iWT2_2, with the exception of the iK381X iPS cell line, which expresses a truncating heterozygous variant (Figs 1B, 3A and 3B). In order to investigate the consequences of this variant for BRCA1 protein expression, we generated a K381X BRCA1 mutant construct by site-directed mutagenesis, and Western blot analysis revealed the expression of the K381X BRCA1 truncated protein in 293T cells (Fig 3C).

We next examined the ability of BRCA1 variant cells to form foci following induction of DNA double-strand breaks by ionising radiation treatment. The expression of BRCA1 variants studied here did not affect neither the formation nor the number of BRCA1 (Fig 3D) and γH2AX foci (S12 Fig).

## Additional variants present in heterozygous BRCA1 variant carriers

Heterozygous *BRCA1* variants can co-occur with other genetic mutations, for instance in TP53 [24, 25]. Our whole exome sequencing data were then inspected for the presence of additional variants. Interestingly, only one known pathogenic variant–a mutation in the ATM gene (c.C8373A:p.Y2791X)—was found in the non-pathogenic Y856H fibroblasts and iPS cells (Fig 4A and 4B). This variant leads to a truncation in the kinase domain of ATM, thus abrogating its autophosphorylation on Ser1981, which is a prerequisite of its function at DNA damage sites [26]. Here, despite the low quality of our Western blot, no ATM autophosphorylation was observed in the iY856H_2 iPS cells compared to WT lines (Fig 4C). Importantly, the Y856H BRCA1 variant is classified as benign (Fig 1B). The presence of this additional variant in ATM gene, therefore, may explain, at least partially, the strong family history of cancer (S2B Fig).

## Reduced DNA double-strand break repair via HR in heterozygous BRCA1 variant carriers

Next, the proliferation rate of BRCA1 variants was assessed. While in the absence of treatment the rate was similar in all four genotypes tested, as expected the two known pathogenic variants, C61G and K381X, were hypersensitive to ionising radiation (IR) treatment (Fig 5A) [27]. Interestingly, the iY856H_2 line, despite an ineffective ATM protein, did not present any hypersensitivity to ionising radiation (IR) treatment, and responded in a similar way to the two WT lines tested (Fig 5A).

In order to determine the consequences of this type of damage at a molecular level, γH2AX, a well-established marker of DNA double-strand breaks, was analysed to monitor the level of damage, and the kinetics of DNA damage repair following exposure to ionising radiation [28–32]. Historically, analyses of γH2AX foci formation and clearance have been performed in lymphoblastoid cell lines to ascertain the impact of BRCA1 variant expression and

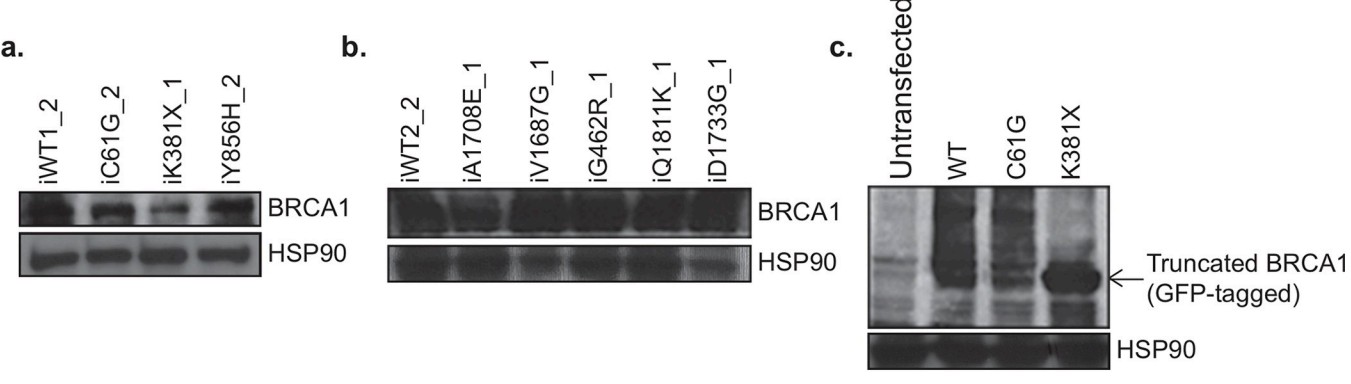

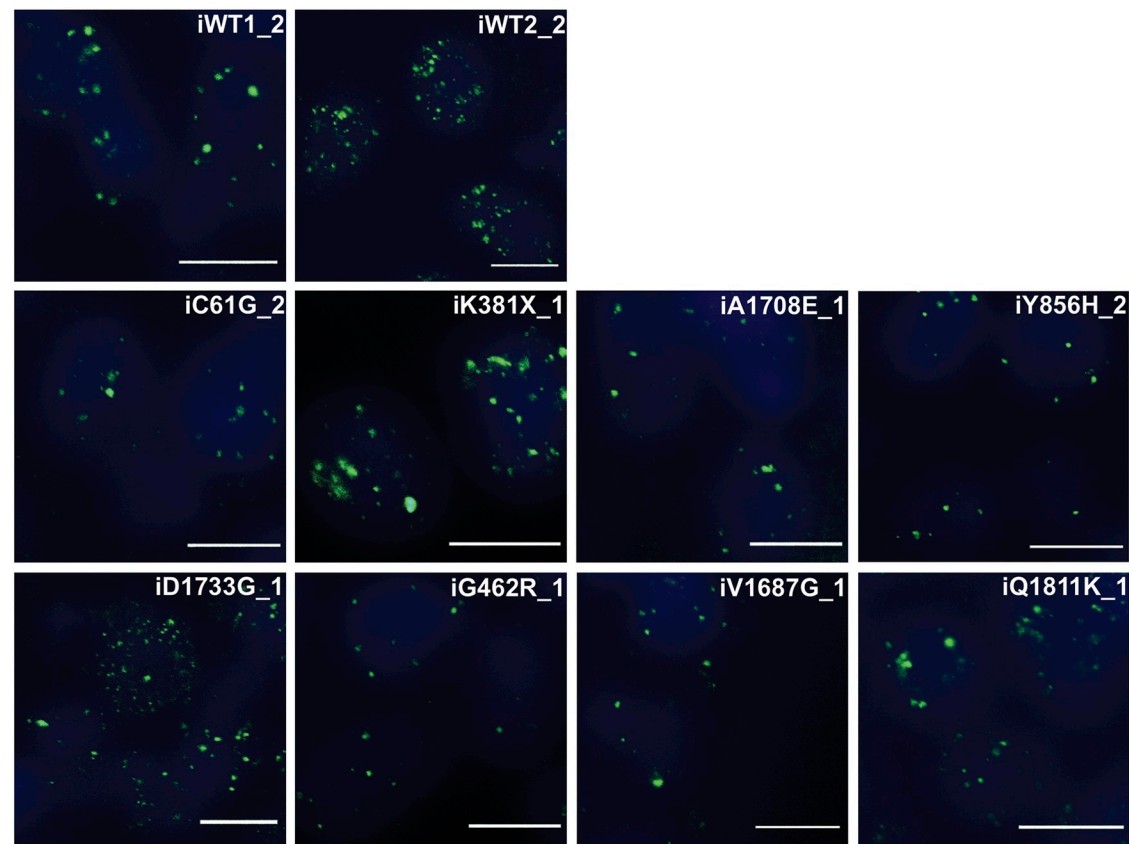

**Fig 3. Assessment of BRCA1 protein expression level in iPS cells derived from fibroblasts. a, b** Expression levels demonstrated by Western blot of whole-cell lysates extracted from iPS cell lines. HSP90 was used as the loading control. **c** 293T cells were transfected with GFP-BRCA1-WT or variants. The truncated form of BRCA1 is indicated by an arrow. HSP90 was used as the loading control. **d** Immunofluorescence staining for BRCA1 nuclear foci formation following ionising radiation exposure. Nuclei were counterstained with DAPI (blue). Scale bars represent 10 μm.

heterozygous status for BRCA1 and BRCA2 on sensitivity to radiation [33–35]. Here, a significant increase in γH2AX foci was observed in all cell lines, which had been exposed to ionising radiation 1 hour previously, compared to untreated cells (Fig 5B–5D, S13 Fig) [36]. However, this induction was significantly lower in the two pathogenic lines, iC61G_2 and iK381X_1, when compared to iWT1_2 and iWT2_2 control lines (respectively 44% and 43% versus 74%

**a.**

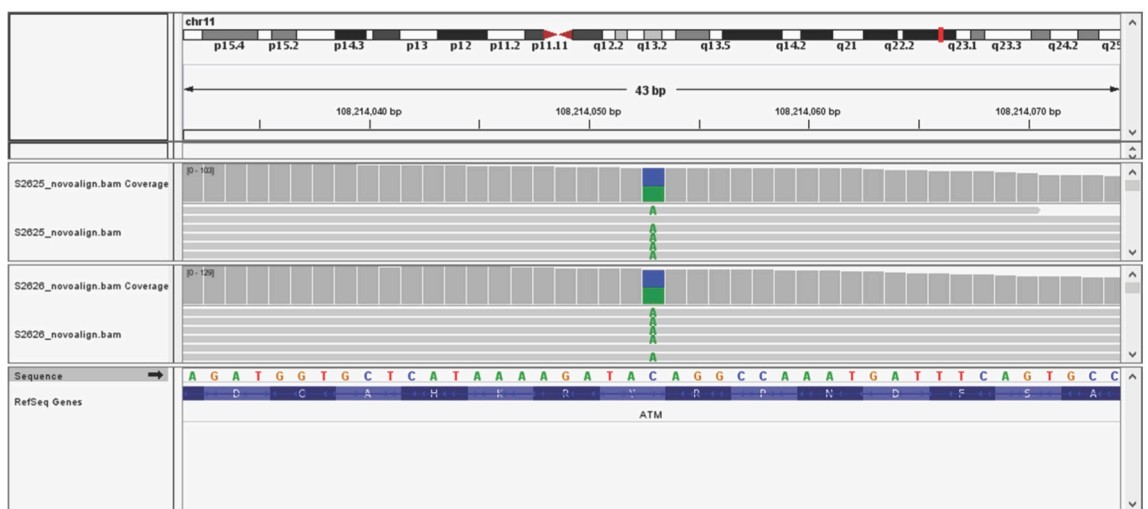

**b.**

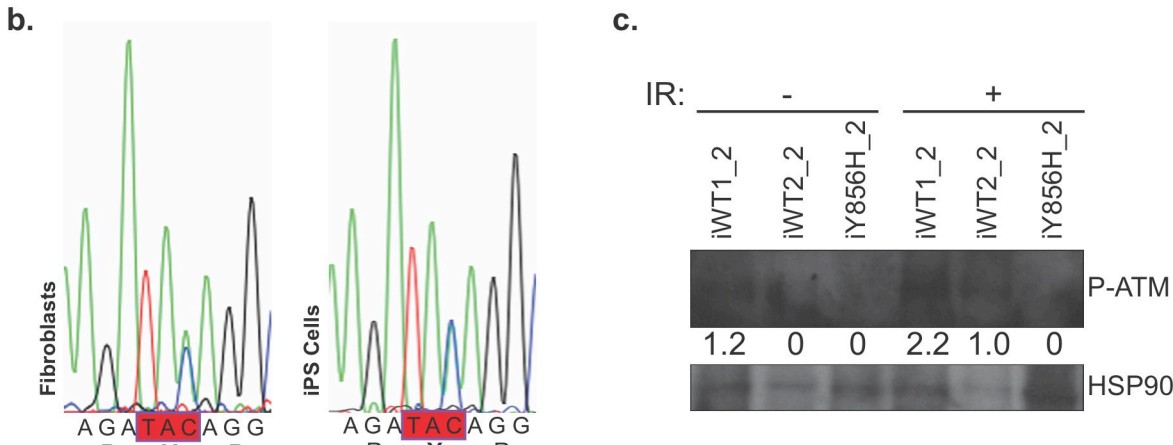

**c.**

**Fig 4. Whole exome sequencing analysis identifies a known pathogenic truncating variant in ATM in the Y856H patient. a** Screenshot of IGV. Track 1 shows mapped reads from iY856H. Track 2 shows mapped reads from matched fibroblasts. The green As in the grey alignments represent bases that did not match the reference sequence. **b** Validation by Sanger sequencing, in fibroblast and in iPS cells, of the heterozygous variant in the ATM gene identified by WES. **c** iPS cells were untreated, or treated by exposure to 0.5 Gy of ionising radiation (IR). Whole-cell lysates were prepared 1 hour post-IR, and analysed by Western blot for ATM phosphorylation at Serine 1981. HSP90 was used as the loading control. ATM autophosphorylation levels were measured using ImageJ quantification tool, and the HSP90 normalized intensity values are presented under the blots.

and 68%; Fig 5D), indicating that the repair response had become corrupted. Also, surprisingly, no variation was observed between the two WT controls and the iY856H_2 line. Nonetheless, all genotypes were capable of clearing γH2AX foci by 12 hours post-treatment, as shown by a dramatic reduction in the number of nuclear foci present, which returned to the untreated basal level (Fig 5C and 5D).

Since BRCA1 plays an essential role in the maintenance of genomic stability via the Homologous Recombination (HR) pathway, HR repair assay is now the standard predictive tool employed in the functional characterisation of BRCA1 variants [7, 37]. Here, in order to evaluate the efficiency of the HR pathway in our eight different variant lines, a well-established reporter system was used [18, 32]. The efficiency of repair via HR in the two WT iPS cell lines

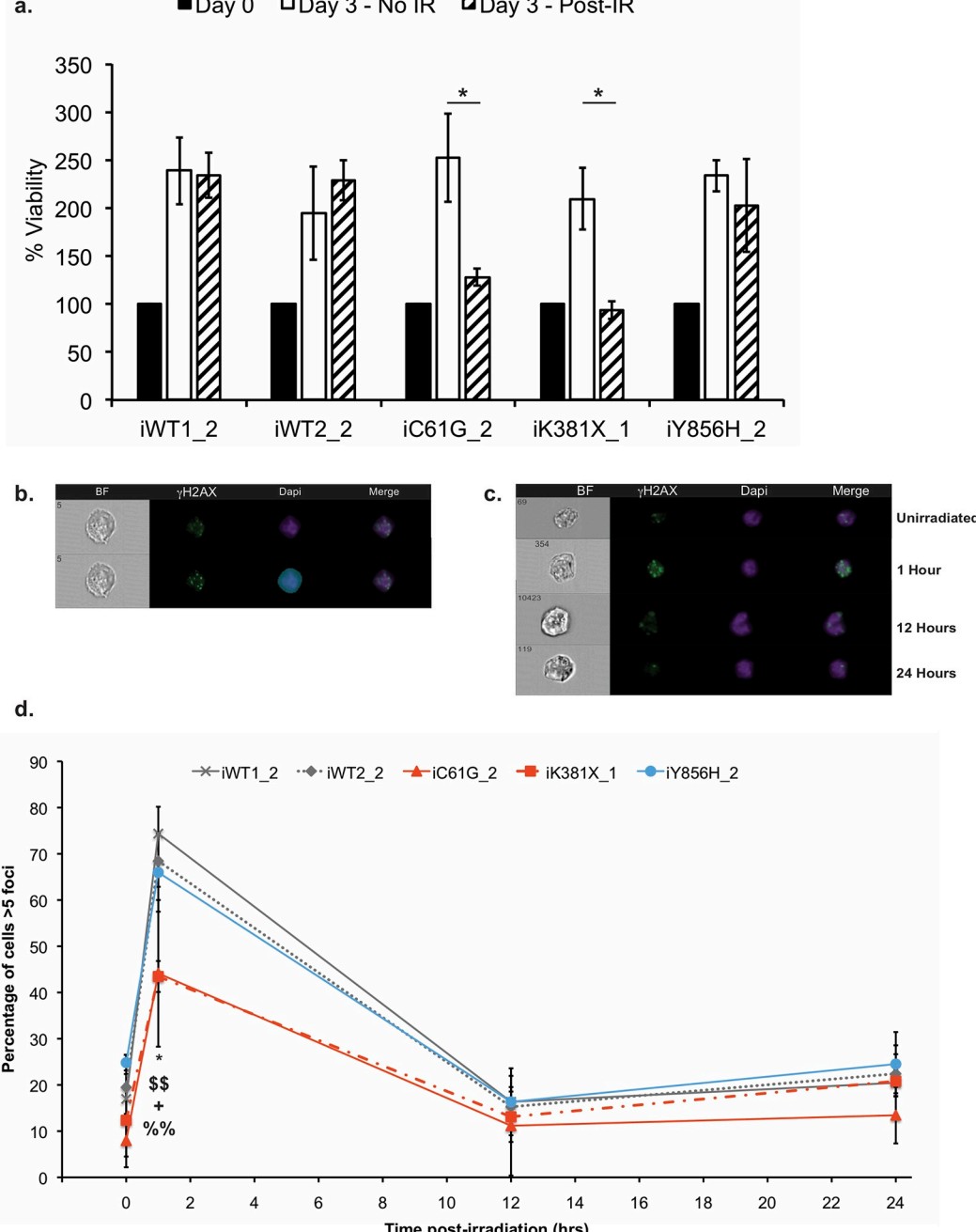

**Fig 5. Impact of BRCA1 variants on response to DNA damage repair. a** Cell viability assay of WT and indicated BRCA1 variant iPS cell lines following IR treatment. The viable fraction is expressed as a percentage of the viability values obtained for the respective untreated genotypes. (*) indicates p<0.05 using Student's t-test. **b** The mask defining strategy for ImageStreamX. The upper panel shows raw data before the application of feature masks for the analysis. The lower panel shows feature masks in cyan. BF = brightfield. Magnification = 40x. **c** Representative images of iPS cells following IR exposure for the time points indicated. **d** All lines have been analysed prior to, and at 1, 12, and 24 hours post-irradiation. The percentage of cells with >5 foci has been plotted. (*) indicates significant difference (p = 0.02) between iWT1_2 and iC61G_2, (§§) indicates significant difference (p = 0.003) between iWT1_2 and iK381X_1, (##) indicates significant difference (p = 0.005) between iWT2_2 and iC61G_2, and (§§) indicates significant difference (p = 0.048) between iWT2_2 and iK381X_1 using Student's t-test. No significant difference was observed between iY856H_2 and the two WT lines analysed. Error bars represent standard error of the mean. N = 3 for iWT1_2, iWT2_2, iK381X_1 and iY856H_2, N = 4 for iC61G_2.

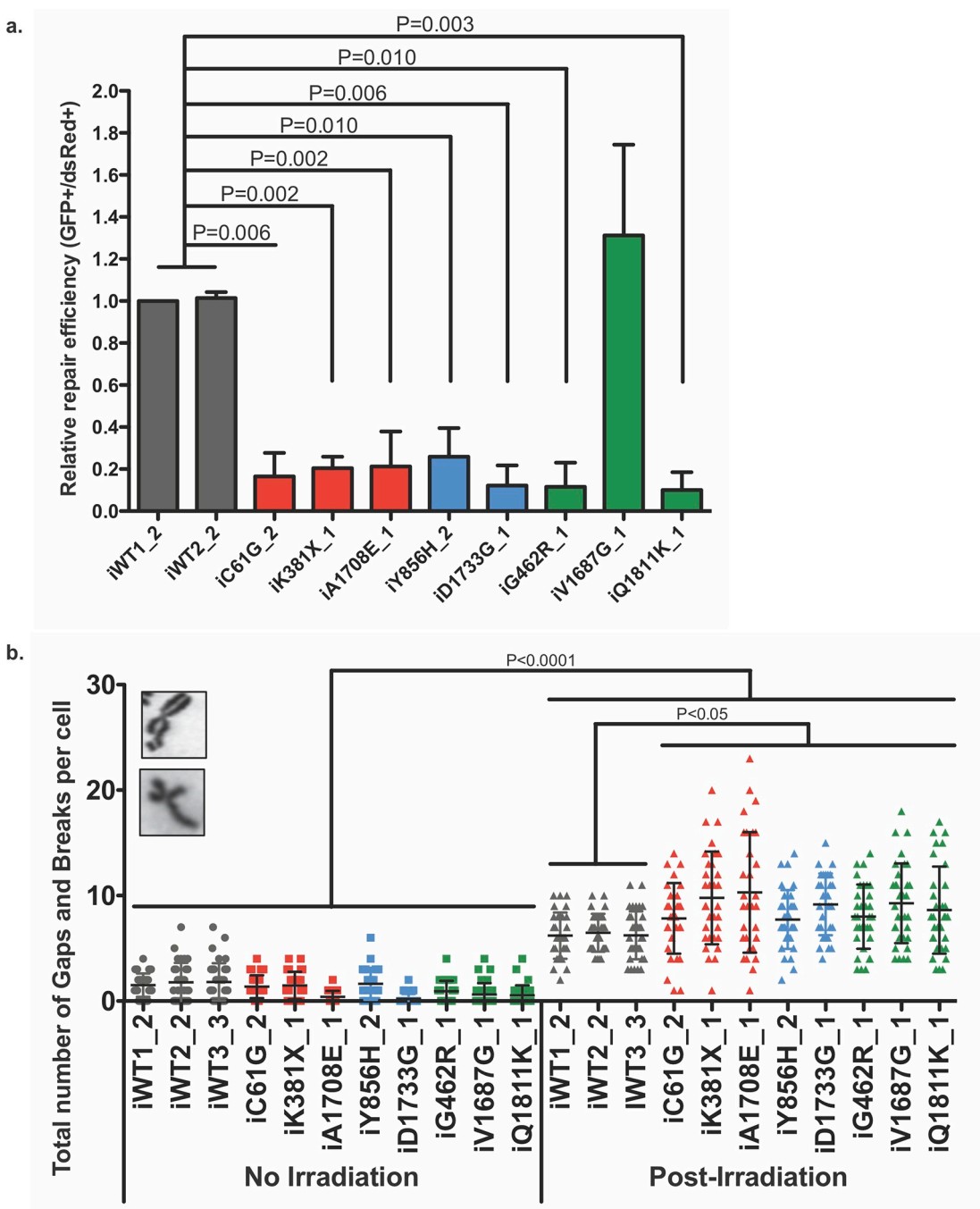

**Fig 6. Effect of BRCA1 variants on HR pathway efficiency. a** The efficiency of repair was measured by quantification of GFP fluorescence expression, which can only occur when the linearized plasmid is accurately repaired. Ratios of GFP+ to DsRed+ were normalised to iWT1_2 iPS cells. Student's t-test was used to measure the significance; p-values are shown on the graph. N≤3. **b** Number of gaps and breaks per metaphase spread of each iPS cell line, with or without ionising radiation treatment. Error bars represent standard error of the mean. Statistical significance was determined using a Mann-Whitney U test. p-values are shown on the graph. N = 30 metaphase cells for each data point. Representative images (inset) of a chromatid gap and a chromatid break.

tested, iWT1_2 and iWT2_2 (Fig 6A) showed a similar degree of efficiency. Analysis of the three known pathogenic lines, iC61G_2, iK381X_1 and iA1708E_1, revealed a significant reduction in the efficiency of repair compared to the two WT controls (Fig 6A). Importantly,

HR repair activity in 4 out of 5 variants (iG462R_1, iY856H_2, iD1733G_1 and iQ1811K_1) tested was also highly inefficient when compared to WT controls (Fig 6A). Interestingly, VUS iV1687G_1 did not show any deficiency in double-strand break repair via HR (Fig 6A).

Chromosomal aberrations are characteristic features of a deficient HR pathway, and historically, were the standard feature analysed to detect the pathogenic effects of heterozygous BRCA1 variants [34, 38–42]. Accordingly, all iPS cells were irradiated, and gaps and breaks were quantified (Fig 6B). As expected, all genotypes presented a similar number of aberrations before treatment, and a significantly higher number of aberrations in cells exposed to ionising radiation compared to untreated cells (Fig 6B). While no difference was observed between the three WT lines tested, all pathogenic, non-pathogenic and VUS lines compared with them exhibited significantly more aberrations (Fig 6B).

## BRCA1 iPS cells maintain their differentiation potential

The differentiation potential of iPS cells represents one of the many advantages of using them as a disease model [11, 43, 44]. Since BRCA1 has been shown to affect mammary differentiation [45–47], we investigated whether the two known pathogenic variants and the Y856H non-pathogenic variant could form non-adherent mammospheres under optimal conditions. As shown in Fig 7A, all genotypes were able to differentiate into primary mammospheres without any obvious variation in the number and size of spheres compared to the two WT lines, and primary mammospheres from the two WT lines and the three variants tested gave rise to secondary and tertiary mammospheres, indicating the presence of self-renewing cells. Moreover, the vast majority of cells forming mammospheres expressed the stem cell markers Nestin and OCT-4 (Fig 7B).

Next, mammospheres were tested to determine whether they could be further differentiated into luminal and basal lineages. Mammospheres disassociated into single cells were then cultured in differentiation conditions, and stained for a luminal epithelial-specific marker, Cytokeratin 18 (CK18), and two basal epithelial-specific markers, Cytokeratin 14 (CK14) and alpha-SMA (smooth muscle actin) [48, 49]. Expression of BRCA1 variants did not alter the capacity of mammospheres to differentiate into both lineages, as shown in Fig 7C (example of iC61G_2).

## Discussion

In this study, we investigated a number of iPS cell lines carrying different heterozygous BRCA1 variants, including three known pathogenic variants, two non-pathogenic variants and three VUSs. All fibroblasts isolated from patients were reprogrammed successfully, and none of the eight BRCA1 variants examined affected the attainment of pluripotency, or the early development of the reprogrammed cells, which is in accordance with previous studies [50, 51].

Since variant classification has important clinical implications, it clearly merits substantial investigation. We speculated that iPS cells might represent an important means of standardising functional studies of BRCA1 variants. Indeed, all these previous studies are frequently ambivalent, yielding different results depending on the cellular model used, and/or the functional assay tested. More importantly, to date BRCA1 variants have been studied in isolation i.e. without consideration of other genetic aberrations present in the patient. However, the modern approach to cancer treatment requires a full understanding of the cancer phenotype if adequate treatment is to be provided [52]. Because only BRCA1 status was recorded for the fibroblast samples provided by our clinical collaborators, we therefore performed WES analysis on all the iPS cell lines we generated, in order to characterise them further. Through this

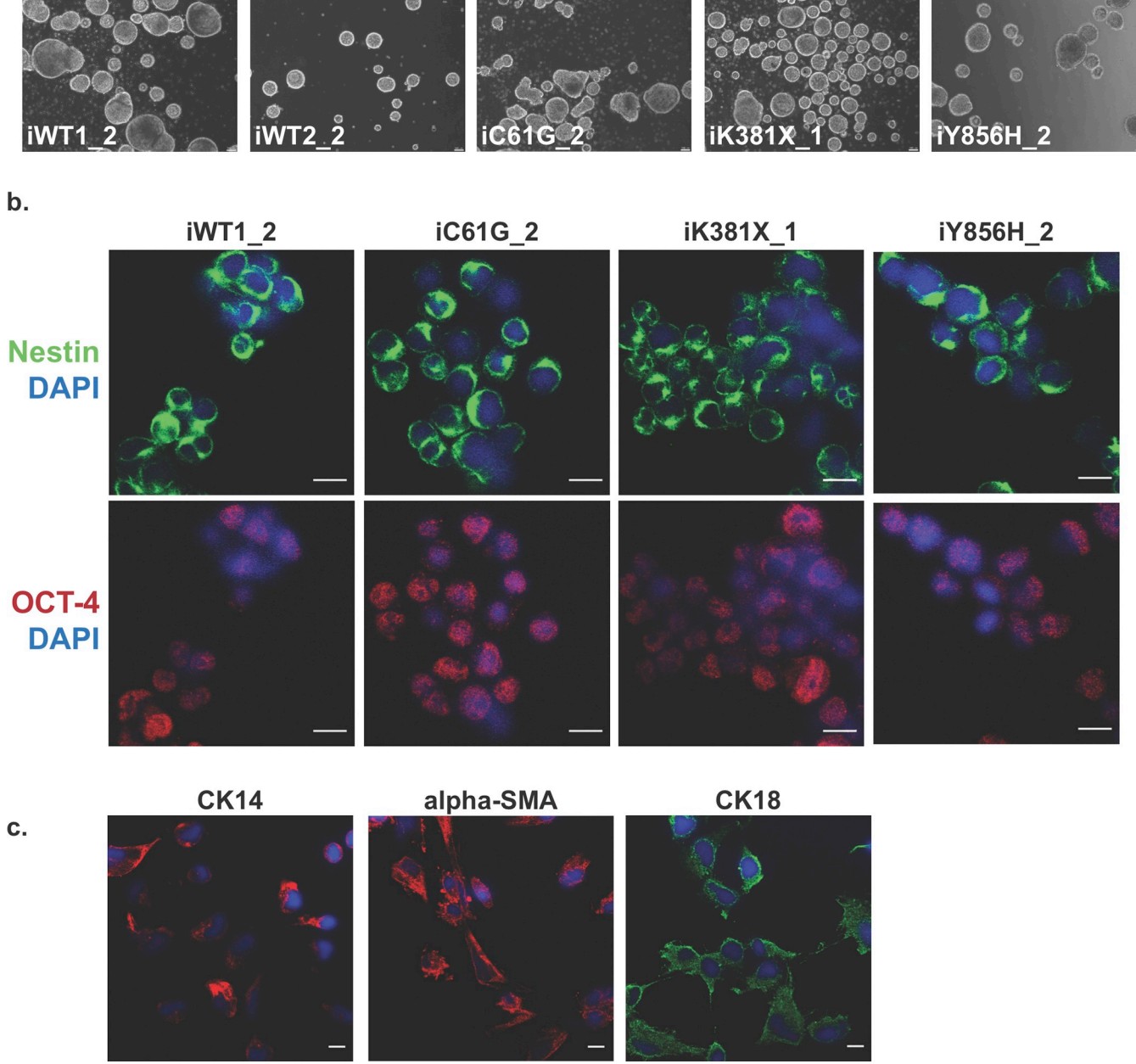

**Fig 7. Differentiation of induced pluripotent stem cells into mammospheres. a** Representative images of iWT1_2, iWT2_2, iC61G_2, iK381X_1 and iY856H_2 iPS cells differentiated into primary mammospheres using MammoCult medium. **b** Representative images of iPS cells-derived mammospheres stained for two stem markers: Nestin (green) and OCT-4 (red). Nuclei were counterstained with DAPI (blue). Scale bars represent 10 μm. **c** Immunofluorescence staining for basal (CK14 and alpha-SMA) and luminal (CK18) markers on differentiated mammospheres. Nuclei were counterstained with DAPI (blue). Scale bars represent 10 μm.

analysis, we identified one known pathogenic variant in the Y856H lines. This truncating variant was found in the ATM gene, abolishing its kinase activity [53, 54], and thus probably explaining the strong family history of cancer reported by the patient in question, who carried a classified benign BRCA1 variant [55]. The identification of the additional variant in this

patient stresses the importance of establishing, a defined molecular signature; the ENIGMA consortium (Evidence-based Network for the Interpretation of Germline Mutant Alleles), for example, is currently working on this [52, 56–58].

Locus-specific loss of heterozygosity (LOH) (i.e. absence of wild-type BRCA1 copy) is observed in BRCA1 breast cancers [59]; however, the impact of LOH on BRCA1 variants are still under investigation [60]. All our iPS cell lines were generated from heterozygous BRCA1 variant carriers, and no loss of the wild-type copy in culture was observed (possibly because we never used cell cultures higher than passage P15). The potential impact of BRCA1 LOH on BRCA1 VUS effects is of great interest as these results could drastically change the designation of some VUSs [59]. Moreover, iPS cells could then be an adequate cellular model to study the emergence of LOH, and help establishing whether BRCA1 LOH could be due to the extensive genomic instability induced by BRCA1 variant expression.

The three known pathogenic variants studied here, C61G [20], K381X and A1708E [21, 61–63] presented reduced HR effectiveness and increased level of chromosomal aberrations, reflecting strong defects in DNA repair, as expected. Therefore, we validated these two assays to evaluate the four other variants (likely benign and VUS). Of the four other variants studied, three were functionally impaired, with defective repair via the HR pathway as well as accumulation of chromosome aberrations. The G462R variant was classified as likely to be pathogenic, based on the level of conservation of the residue affected [64], and here we demonstrated its inability to reliably repair DNA damage through the HR pathway. Moreover, the D1733G variant, which was classified as likely to be benign, based on two studies of VUS characterisation in yeast [65, 66] and on a saturation genome editing study [67], was found by us to be functionally impaired. This was also the case for the Q1811K variant, which had previously been considered non-functional [65]. Only one VUS, V1687G, presented inconsistency between the two assays performed. In the case of D1733G and V1687G, a deeper analysis of WES data would be of interest, as VUSs might be found in other breast-cancer-related genes, thus explaining these discrepant results. Thus, likewise genetic testing alone is unsatisfactory for the classification of BRCA1 VUSs, functional assays appear also insufficient regardless of the cellular model used: establishing a reliable classification of variants will only be possible by integrating data from multiple sources such as family history, gene and protein structure and functional assays [52, 58, 68].

A central advantage of iPS cell technology is the ability of iPS lines to differentiate into multiple lineages depending on the specific culture conditions adopted. This presents breast cancer researchers with a golden opportunity to study their capacity to differentiate into multiple lineages, or to exhibit a tendency towards differentiation into one particular lineage. Here, we demonstrated the ability of all iPS cell lines tested to differentiate into mammospheres under suitable culture conditions, and have shown that these mammospheres in turn generated both basal and luminal epithelial lineages. These data should therefore encourage the initiation of new research studies designed to investigate further the differentiation potential of mammospheres derived from BRCA1 variants (i.e. qualitative analysis of luminal versus basal differentiation), their subsequent ability to engraft into mammary fat pads, and their tumorigenic potential. In this way, iPS cells may constitute an important new tool for the better understanding of the pathophysiology and clinical evolution of breast cancer.

## Supporting information

**S1 Fig. Complete loss of reprogramming factors and Sendai virus sequences in all iPS cells.**
Taqman qPCR analysis of reprogramming factor and SeV sequences in **a** iWT1_2. **b** iG462R_1. **c** iC61G_2. **d** iD1733G_1. **e** iK381X_1. **f** iV1687G_1. **g** iY856H_2. **h** iQ1811K_1. **i**

iA1708E_1 iPS cells compared to uninfected fibroblasts.
(ZIP)

**S2 Fig. Pedigrees. a** The six-generation pedigree of the BRCA1 G462R family. **b** The three-generation pedigree of the BRCA1 Y856H family. **c** The four-generation pedigree of the BRCA1 D1733G family. **d** The four-generation pedigree of the BRCA1 Q1811K family. **e** The five-generation pedigree of the BRCA1 V1687G family. Patient ages at the time of cancer occurrence are located to the upper left of each symbol. Symbols coloured blue or purple indicate patients with breast cancer or ovarian cancer, respectively; pink indicates skin cancer, green indicates lung cancer, orange indicates prostate cancer, dark red indicates leukaemia, and grey denotes cancers with an unknown primary site. A diagonal indicates deceased individuals. Numbers inside symbols indicate multiple individuals. Asterisks (*) identify patients whose biopsies were used for iPS cell derivation. Nk = age at the time of cancer not known.
(ZIP)

**S3 Fig. No impact of BRCA1 variants on the generation of iPS cells.** No variation of efficiency to generate induced pluripotent stem cell (iPSC) colonies was observed between all iPS cell lines.
(TIFF)

**S4 Fig. Characterisation of iWT1_2 and iC61G_2 iPS cell line. a** Representative staining for pluripotency markers for iWT1_2. Nuclei were counterstained with DAPI (blue). Scale bars represent 100 μm. **b** Representative histological analysis of hematoxylin-eosin-stained images of sections of teratomas derived from iWT1_2 cells, showing all three germ-layers labelled (1 = endoderm, 2 = mesoderm, 3 = ectoderm). **c** Sanger sequencing showing WT sequence, heterozygous C61G variant present in fibroblasts, and in iPS cells.
(TIFF)

**S5 Fig. Characterisation of iK381X_1 iPS cell line. a** Representative staining for pluripotency markers. Nuclei were counterstained with DAPI (blue). Scale bars represent 100 μm. **b** Representative histological analysis of hematoxylin-eosin-stained images of sections of teratomas derived from iPS cells, showing all three germ-layers labelled (1 = endoderm, 2 = mesoderm, 3 = ectoderm). **c** Sanger sequencing showing WT sequence, heterozygous K381X variant present in fibroblasts, and in iPS cells.
(TIFF)

**S6 Fig. Characterisation of iY856H_2 iPS cell line. a** Representative staining for pluripotency markers. Nuclei were counterstained with DAPI (blue). Scale bars represent 100 μm. **b** Representative histological analysis of hematoxylin-eosin-stained images of sections of teratomas derived from iPS cells, showing all three germ-layers labelled (1 = endoderm, 2 = mesoderm, 3 = ectoderm). **c** Sanger sequencing showing WT sequence, heterozygous Y856H variant present in fibroblasts, and in iPS cells.
(TIFF)

**S7 Fig. Characterisation of iA1708E_1 IPS cell line. a** Representative staining for pluripotency markers. Nuclei were counterstained with DAPI (blue). Scale bars represent 100 μm. **b** Representative staining of in vitro differentiation potential of iPS cells using specific antibodies against the endodermal marker α-Feto Protein, ectodermal marker β III Tubulin and mesodermal markers α-smooth muscle actin (SMA). Nuclei were counterstained with DAPI (blue). Scale bars represent 100 μm. **c** Sanger sequencing showing WT sequence, heterozygous A1708E variant present in fibroblasts, and in iPS cells.
(TIFF)

**S8 Fig. Characterisation of iG462R_1 iPS cell line. a** Representative staining for pluripotency markers. Nuclei were counterstained with DAPI (blue). Scale bars represent 100 μm. **b** Representative staining of in vitro differentiation potential of iPS cells using specific antibodies against the endodermal marker α-Feto Protein, ectodermal marker β III Tubulin and mesodermal markers α-smooth muscle actin (SMA). Nuclei were counterstained with DAPI (blue). Scale bars represent 100 μm. **c** Sanger sequencing showing WT sequence, heterozygous G462R variant present in fibroblasts, and in iPS cells.
(TIFF)

**S9 Fig. Characterisation of iV1687G_1 iPS cell line. a** Representative staining for pluripotency markers. Nuclei were counterstained with DAPI (blue). Scale bars represent 100 μm. **b** Representative staining of in vitro differentiation potential of iPS cells using specific antibodies against the endodermal marker α-Feto Protein, ectodermal marker β III Tubulin and mesodermal markers α-smooth muscle actin (SMA). Nuclei were counterstained with DAPI (blue). Scale bars represent 100 μm. **c** Sanger sequencing showing WT sequence, heterozygous V1687G variant present in fibroblasts, and in iPS cells.
(TIFF)

**S10 Fig. Characterisation of iQ1811K_1 iPS cell line. a** Representative staining for pluripotency markers. Nuclei were counterstained with DAPI (blue). Scale bars represent 100 μm. **b** Representative staining of in vitro differentiation potential of iPS cells using specific antibodies against the endodermal marker α-Feto Protein, ectodermal marker β III Tubulin and mesodermal markers α-smooth muscle actin (SMA). Nuclei were counterstained with DAPI (blue). Scale bars represent 100 μm. **c** Sanger sequencing showing WT sequence, heterozygous Q1811K variant present in fibroblasts, and in iPS cells.
(TIFF)

**S11 Fig. Characterisation of iD1733G_1 iPS cell line. a** Representative staining for pluripotency markers. Nuclei were counterstained with DAPI (blue). Scale bars represent 100 μm. **b** Representative staining of in vitro differentiation potential of iPS cells using specific antibodies against the endodermal marker α-Feto Protein, ectodermal marker β III Tubulin and mesodermal markers α-smooth muscle actin (SMA). Nuclei were counterstained with DAPI (blue). Scale bars represent 100 μm. **c** Sanger sequencing showing WT sequence, heterozygous D1733G variant present in fibroblasts, and in iPS cells.
(TIFF)

**S12 Fig. Assessment of γH2AX protein expression level in iPS cells derived from fibroblasts.** Immunofluorescence staining for γH2AX nuclear foci formation following ionising radiation exposure. Nuclei were counterstained with DAPI (blue). Scale bars represent 10 μm.
(TIFF)

**S13 Fig. Quantification of γH2AX foci by ImageStreamX.** Representative foci quantification graphs for iWT1_2, iWT2_2, iC61G_2, iK381X_1 and iY856H_2 iPS cells before and after IR treatment.
(TIFF)

**S1 Table. List of primers used in this study.**
(XLSX)

**S1 File. Whole exome sequencing results.**
(XLSX)

**S1 Raw images.**
(TIFF)

## Acknowledgments

The authors sincerely thank all those patients, and their families, who have generously donated time, samples, and information. The authors are grateful for the collaboration of the breast and plastic surgical teams, breast tissue bank co-ordinators, led by Cheryl Gillett, and the diagnostic DNA laboratory. They also wish to acknowledge their deep gratitude to Dusko Ilic and his group, for their help and support throughout the project. The authors are indebted to all members of the Biomedical Research Centre (BRC) Flow Cytometry Core Facility, and particularly to Pj Chana and Susanne Heck. The authors thank Alka Saxena (former Director, BRC Genomics Core Facility) for her support in whole-exome sequencing analysis. The authors also thank Davide Danovi and Fiona Watt for giving us access to their HipSci bank. Finally, the authors would like to thank Elizabeth Manners for editing the manuscript.

## Author Contributions

**Conceptualization:** Edwige Voisset, Ellen Solomon.

**Funding acquisition:** Meryem Ozgencil, Edwige Voisset, Ellen Solomon.

**Investigation:** Meryem Ozgencil, Ian Kesterton, Michael Simpson, Paul Sharpe, Edwige Voisset.

**Methodology:** Meryem Ozgencil, Edwige Voisset.

**Project administration:** Edwige Voisset.

**Resources:** Julian Barwell, Marc Tischkowitz, Louise Izatt, Michael Simpson, Paul Sharpe, Paulo de Sepulveda, Edwige Voisset.

**Supervision:** Edwige Voisset.

**Validation:** Meryem Ozgencil, Edwige Voisset.

**Visualization:** Edwige Voisset.

**Writing – original draft:** Meryem Ozgencil, Edwige Voisset.

**Writing – review & editing:** Meryem Ozgencil, Julian Barwell, Marc Tischkowitz, Louise Izatt, Ian Kesterton, Michael Simpson, Paul Sharpe, Paulo de Sepulveda, Edwige Voisset, Ellen Solomon.

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
