## [Decision Letter · Decision Letter 0]

6 Oct 2021

PONE-D-21-28014Assessing BRCA1 activity in DNA damage repair using human induced Pluripotent Stem Cells to assist classification of BRCA1 variants of uncertain significancePLOS ONE

Dear Dr. Voisset,

Thank you for submitting your manuscript to PLOS ONE. After careful consideration, we feel that it has merit but does not fully meet PLOS ONE’s publication criteria as it currently stands. Therefore, we invite you to submit a revised version of the manuscript that addresses the points raised during the review process.

We look forward to receiving your revised manuscript.

Kind regards,

Alvaro Galli

Academic Editor

PLOS ONE

Journal Requirements:

4. Please upload a copy of Supporting Information Figures S1-S10 which you refer to in your text on pages 13, 15, 17.

Reviewers' comments:

Reviewer's Responses to Questions

**Comments to the Author**

1. Is the manuscript technically sound, and do the data support the conclusions?

Reviewer #1: Yes

2. Has the statistical analysis been performed appropriately and rigorously? 

Reviewer #1: Yes

3. Have the authors made all data underlying the findings in their manuscript fully available?

Reviewer #1: Yes

4. Is the manuscript presented in an intelligible fashion and written in standard English?

Reviewer #1: Yes

5. Review Comments to the Author

Reviewer #1: In this manuscript Ozgencil et al. present an analysis of a subset of BRCA1 variants in a new model of induced pluripotent cells that can form three-dimensional mammospheres. The authors argue that this system constitute an important cell context to evaluate the function of BRCA1 variants. The authors isolate fibroblasts from carriers, induce pluripotency in vitro, and then characterize and analyze the behavior of heterozygous cell lines in their ability to form pluripotent colonies, and in their response to DNA damage. The paper is well written, the results are of high quality and the analysis of variants of uncertain significance is timely.

There are a few issues that the authors should consider revising.

While I agree that this model is extremely important to add to the analysis of unclassified variants and there are several advantages of this model over existing ones, the claim in the title “to assist classification of BRCA1 variants of uncertain significance” maybe premature. I believe the authors can say that it may be a powerful method to do so, and argue that point in the paper, but at this stage having this in the title suggests that the assay has been validated, which is not the case (need for additional controls to correctly calibrate the assay, for example).

The issue of haploinsufficiency of BRCA1 is far from being resolved and whether the presence of a wild-type copy of BRCA1 in these cells affects the results. For example, whether a BRCA1 variant that has no detectable effect in heterozygosis (with a wild type copy), would show a defect when in hemizygozis after the loss of the wt copy. Are there dominant negative effects of any variant? (why does the cell with transfected K381X does not appear to have the wt copy?) The issue of the heterozygote context should be discussed in more detail and not mentioned only in passing.

The authors claim that “The three known pathogenic variants studied here, C61G [20], K381X and A1708E 554 [21, 59-61], were correctly classified according to our evaluation.” Needs to be clarified. The authors conduct several assays, and the ‘correct’ classification seems dependent on what do the authors refer to their ‘evaluation’. While the three pathogenic variants mentioned score as loss of function in the relative repair efficiency assay, they do not differ from the wt in the ability to form mammosphere, or expression of markers. The A1708E is not tested in any downstream assays. The authors should define what do they mean by their evaluation.

I think the following sentence, which may be referring to explaining the family history, is excessively broad if the authors mean for classifying a variant. Please clarify.

“Thus, likewise genetic testing alone is unsatisfactory for the classification of BRCA1 568 VUSs, functional assays are also insufficient regardless of the cellular model used 569 [52, 58].”

Finally, I wonder if the authors are selling themselves short on the last assay (ability to form luminal and basal lineage) where they report that there is no qualitative difference. Given the role in BRCA1 in shifting basal/luminal differentiation, I wonder if the authors have looked at it quantitatively, before discounting it as a way to assist in the classification of variants.

6. PLOS authors have the option to publish the peer review history of their article (what does this mean?). If published, this will include your full peer review and any attached files.

Reviewer #1: No

---

## [Author Response · Author response to Decision Letter 0]

8 Nov 2021

RESPONSES TO THE EDITOR’ COMMENTS

Our detailed responses are below, and the highlighted copy of our manuscript shows the changes (in dark red) we have made in response to all comments received.

Moreover, in order to fulfil Plos One criteria, we added a supporting information file with uncropped and unadjusted Western-blot images (S1 Raw data), we also added all figures and files that were originally stated as “data not shown”, and renamed our files as requested.

We hope that you will find this revised manuscript suitable for publication in Plos One.

RESPONSES TO THE REVIEWER’ COMMENTS

We have made the changes and amendments suggested by the Reviewer. Our responses to the individual comments are shown below.

We hope that the Reviewer will find our responses satisfactory, and that our manuscript will now prove acceptable for publication in Plos One.

Review Comments to the Author

Reviewer #1: In this manuscript Ozgencil et al. present an analysis of a subset of BRCA1 variants in a new model of induced pluripotent cells that can form three-dimensional mammospheres. The authors argue that this system constitutes an important cell context to evaluate the function of BRCA1 variants. The authors isolate fibroblasts from carriers, induce pluripotency in vitro, and then characterize and analyze the behavior of heterozygous cell lines in their ability to form pluripotent colonies, and in their response to DNA damage. The paper is well written, the results are of high quality and the analysis of variants of uncertain significance is timely.

There are a few issues that the authors should consider revising.

Point 1: While I agree that this model is extremely important to add to the analysis of unclassified variants and there are several advantages of this model over existing ones, the claim in the title “to assist classification of BRCA1 variants of uncertain significance” maybe premature. I believe the authors can say that it may be a powerful method to do so, and argue that point in the paper, but at this stage having this in the title suggests that the assay has been validated, which is not the case (need for additional controls to correctly calibrate the assay, for example).

Answer: We understand the point that the Reviewer is making, and have altered the title of the manuscript to “Assessing BRCA1 activity in DNA damage repair using human induced Pluripotent Stem Cells as an approach to assist classification of BRCA1 variants of uncertain significance”.

Point 2: The issue of haploinsufficiency of BRCA1 is far from being resolved and whether the presence of a wild-type copy of BRCA1 in these cells affects the results. For example, whether a BRCA1 variant that has no detectable effect in heterozygosis (with a wild type copy), would show a defect when in hemizygozis after the loss of the wt copy. Are there dominant negative effects of any variant? (why does the cell with transfected K381X does not appear to have the wt copy?) The issue of the heterozygote context should be discussed in more detail and not mentioned only in passing.

Answer: We thank the Reviewer for raising this point on LOH, which is indeed quite critical, and that we did not address since we only had heterozygous variant cell lines. We have now added a section on this topic in the discussion (lines 551-560).

Concerning the K381X variant- this mutation introduces a premature stop codon resulting in a truncated protein. We were unable to detect the endogenous form of this variant. The WT allele can however be observed in Figure 3a, and the signal is significantly reduced compared to the WT control since only one full-length allele is present. A GFP-tagged construct was then transfected in 293T cells (Figure 3c) to determine whether this variant can be expressed. In order to clarify this point, we have specified it on Figure 3c.

Point 3: The authors claim that “The three known pathogenic variants studied here, C61G [20], K381X and A1708E [21, 59-61], were correctly classified according to our evaluation.” Needs to be clarified. The authors conduct several assays, and the ‘correct’ classification seems dependent on what do the authors refer to their ‘evaluation’. While the three pathogenic variants mentioned score as loss of function in the relative repair efficiency assay, they do not differ from the wt in the ability to form mammosphere, or expression of markers. The A1708E is not tested in any downstream assays. The authors should define what do they mean by their evaluation.

Answer: We apologize for the lack of clarity. This sentence has now been corrected as follow:

“The three known pathogenic variants studied here, C61G [20] and K381X and A1708E [21, 61-63] presented reduced HR effectiveness and increased level of chromosomal aberrations, reflecting strong defects in DNA repair, as expected. Therefore, we validated these two assays to evaluate the four other variants (likely benign and VUS).”

Point 4: I think the following sentence, which may be referring to explaining the family history, is excessively broad if the authors mean for classifying a variant. Please clarify.

“Thus, likewise genetic testing alone is unsatisfactory for the classification of BRCA1 VUSs, functional assays are also insufficient regardless of the cellular model used [52, 58].”

Answer: We apologize for the lack of clarity. This sentence has now been corrected as follow:

“Thus, likewise genetic testing alone is unsatisfactory for the classification of BRCA1 VUSs, functional assays appear also insufficient regardless of the cellular model used: establishing a reliable classification of variants will only be possible by integrating data from multiple sources such as family history, gene and protein structure and functional assays [52, 58, 68].”

Point 5: Finally, I wonder if the authors are selling themselves short on the last assay (ability to form luminal and basal lineage) where they report that there is no qualitative difference. Given the role in BRCA1 in shifting basal/luminal differentiation, I wonder if the authors have looked at it quantitatively, before discounting it as a way to assist in the classification of variants.

Answer: We would like to apologize for the misunderstanding: we performed this assay to assess the ability of all BRCA1 variants (1) to form mammospheres and (2) to go through the differentiation process. However, it is a project per se to investigate the degree of luminal versus basal differentiation in all variants, which would require many additional experiments and deep analyses, that we, unfortunately, are unable to afford. We thank the Reviewer for the suggestion, and we surely think this would be the ideal next step of this project.

We have now revised the manuscript to clarify this point (lines 593-594).

---

## [Decision Letter · Decision Letter 1]

18 Nov 2021

Assessing BRCA1 activity in DNA damage repair using human induced Pluripotent Stem Cells as an approach to assist classification of BRCA1 variants of uncertain significance

PONE-D-21-28014R1

Dear Drs. Voisset,

We’re pleased to inform you that your manuscript has been judged scientifically suitable for publication and will be formally accepted for publication once it meets all outstanding technical requirements.

Kind regards,

Alvaro Galli

Academic Editor

PLOS ONE

Additional Editor Comments (optional):

Reviewers' comments:

Reviewer's Responses to Questions

**Comments to the Author**

1. If the authors have adequately addressed your comments raised in a previous round of review and you feel that this manuscript is now acceptable for publication, you may indicate that here to bypass the “Comments to the Author” section, enter your conflict of interest statement in the “Confidential to Editor” section, and submit your "Accept" recommendation.

Reviewer #1: All comments have been addressed

2. Is the manuscript technically sound, and do the data support the conclusions?

Reviewer #1: Yes

3. Has the statistical analysis been performed appropriately and rigorously? 

Reviewer #1: Yes

4. Have the authors made all data underlying the findings in their manuscript fully available?

Reviewer #1: Yes

5. Is the manuscript presented in an intelligible fashion and written in standard English?

Reviewer #1: Yes

6. Review Comments to the Author

Reviewer #1: I am satisfied with the changes provided in the revised manuscript and all concerns were addressed.

7. PLOS authors have the option to publish the peer review history of their article (what does this mean?). If published, this will include your full peer review and any attached files.

Reviewer #1: No

---

## [Editor Report · Acceptance letter]

22 Nov 2021

PONE-D-21-28014R1 

Assessing BRCA1 activity in DNA damage repair using human induced Pluripotent Stem Cells as an approach to assist classification of BRCA1 variants of uncertain significance 

Dear Dr. Voisset:

I'm pleased to inform you that your manuscript has been deemed suitable for publication in PLOS ONE. Congratulations! Your manuscript is now with our production department. 

Kind regards, 

on behalf of

Dr. Alvaro Galli 

Academic Editor

PLOS ONE